

# Robust proportional overlapping analysis for feature selection in binary classification within functional genomic experiments

Muhammad Hamraz[1], Naz Gul[1], Mushtaq Raza[2], Dost Muhammad Khan[1], Umair Khalil[1], Seema Zubair[3] and Zardad Khan[1]

[1] Department of Statistics, Abdul Wali Khan University Mardan, Mardan, Pakistan
[2] Department of Computer Sciences, Abdul Wali Khan University Mardan, Mardan, Pakistan
[3] Department of Mathematics, Statistics and Computer Science, University of Agriculture Peshawar, Peshawar, Pakistan

## ABSTRACT

In this paper, a novel feature selection method called Robust Proportional Overlapping Score (RPOS), for microarray gene expression datasets has been proposed, by utilizing the robust measure of dispersion, i.e., Median Absolute Deviation (MAD). This method robustly identifies the most discriminative genes by considering the overlapping scores of the gene expression values for binary class problems. Genes with a high degree of overlap between classes are discarded and the ones that discriminate between the classes are selected. The results of the proposed method are compared with five state-of-the-art gene selection methods based on classification error, Brier score, and sensitivity, by considering eleven gene expression datasets. Classification of observations for different sets of selected genes by the proposed method is carried out by three different classifiers, i.e., random forest, k-nearest neighbors (k-NN), and support vector machine (SVM). Box-plots and stability scores of the results are also shown in this paper. The results reveal that in most of the cases the proposed method outperforms the other methods.

## INTRODUCTION

Feature or variable selection is the process of selecting a subset of features from a large feature space, especially in high dimensional datasets such as microarray gene expression, for model construction. Selecting a subset of genes/features is a necessary task in classification and regression problems. In regression, the feature or gene selection is carried out to better estimate the average value of the target or response variable, whereas in classification it is used to improve the classification accuracy. The motivation behind feature selection is that there are redundant and/or irrelevant features that do not contribute in regulating the response variable and adversely affect the underlying algorithms. So it is necessary to select those features which are discriminative and can help in simplification of model construction. Moreover, a small number of features help in reducing the training time, increasing the generalizability of the models by minimizing

Corresponding author
Zardad Khan,
zardadkhan@awkum.edu.pk

their variances and reducing the curse of dimensionality in $n < p$ problems. Feature selection can be categorized into three categories, i.e., Wrapper, Embedded and Filter. The details of these methods are given below.

## Wrapper methods

In Wrapper methods, all possible subsets of features in the training set are evaluated by using a predictive model. Each subset is assigned a score based on model accuracy on the hold-out (testing) set. These methods are computationally expensive, since for each feature subset a new predictive model is to be trained. An example of the wrapper method can be found in _Saeys, Inza & Larrañaga (2007)_.

## Embedded methods

These methods are somehow similar to the Wrapper procedures. The embedded feature selection methods differ from the wrapper procedures in the sense that the former do not need to train a new model for each feature subset. In these procedures, gene/feature selection is considered as a constituent of model construction. Some of the most common embedded methods include decision tree algorithm, regression with LASSO and Ridge regression. The last two methods shrink the coefficient of non-informative features to zero and almost zero, respectively. Classification tree based classifier (_Breiman et al., 1984_) is another example of this method.

## Filter methods

In Filter methods, feature selection is carried out by applying a statistical measure such as the mutual information criteria (_Guyon & Elisseeff, 2003_), the pointwise mutual information criteria (_Yang & Pedersen, 1997_) and Pearson product-moment correlation, Relief-based algorithms (_Urbanowicz et al., 2018_), etc., to each feature independently or by finding the association of the feature with the target or response variable. Features are then ranked according to their relevance score. Features with the highest relevance scores are selected for model construction. Other examples of such methods could be seen in _Ghosh et al. (2020)_, _El-Hasnony et al. (2020)_, _Seo & Cho (2020)_, _Algamal & Lee (2019)_.

The proposed method is based on a filtering approach, where the discriminative features or genes that affect the target variable are identified by using the robust measure of dispersion, i.e., median absolute deviation (MAD) for binary class problems. Eleven benchmark gene expression datasets are used to assess the discriminative ability of genes selected by the proposed method. The performance of genes selected through the proposed method is evaluated by using different classifiers, i.e., Random Forest (RF) (_Breiman, 2001_), _K_-Nearest Neighbors (_k_-NN) (_Cover & Hart, 1967_) and Support Vector Machine (SVM) (_Liao, Li & Luo, 2006_).

## RELATED WORK

Feature selection and their utility in classification analyses can be found in several studies. _Dramiński et al. (2008)_ introduced a method called 'relative importance'. In this method, the discriminative genes are identified by constructing a large number of decision trees,

where the genes that mostly contributed to assigning the samples/observations to their true classes are selected. *Ultsch et al. (2009)* proposed a method called 'PUL' in which the informative genes are selected by the help of a measure (PUL-score) based on retrieval information. A method called minimal redundancy maximal relevance (mRMR) was introduced by *Ding & Peng (2005)*, in which genes having maximum relevance with the target class and minimum redundancy are selected. An ensemble version of *Ding & Peng (2005)* named 'mRMRe' was introduced by *De Jay et al. (2013)*. Principal component analysis technique was used by *Lu et al. (2011)*, where those genes are considered informative that corresponded to the component with less variation. A similar study can be found in *Talloen et al. (2007)*, where the factor analysis technique is used rather than principal component analysis. *Ultsch et al., 2009*; *Liu et al. (2013)* compared different feature selection methods in their study. Identification of informative genes by calculating the $p$-value of the statistical tests such as the Wilcoxon rank-sum test and t-test can be found in *Lausen et al. (2004)*. Selection of discriminative genes by exploiting impurity measures, i.e., Gini index, max minority, and information gain can be found in *Su et al. (2003)*. Features or genes can also be selected by analyzing the overlapping degree between the different classes for each gene. A large overlapping degree between the different classes for a particular gene indicates that the gene is non-informative in classifying the observation to their correct class. A study based on the overlapping score of the genes for a binary class problem can be found in *Apiletti et al. (2007)*. This method, named as 'painter's feature selection method' calculates the overlapping degree between the two classes for each gene by considering a single factor, i.e., the size of the overlapping area. Genes that have maximum overlapped regions are assigned higher scores. Genes are then sorted in increasing order based on their scores. This idea was further extended by *Apiletti et al. (2012)*, by taking into account an additional factor, i.e., the number of overlapped observations in the overlapping area for each gene. *Apiletti et al. (2012)* calculated each gene mask by considering the range of training expression intervals, which represents the capability of a gene to correctly classify the observations into their classes without any ambiguity. In this method, a minimum subset of genes that unambiguously assign the maximum number of training samples to their correct classes is identified by considering the gene masks and overlapping scores through the set covering approach. The final subset of discriminative genes is obtained by considering all the genes in the minimum subset and the genes with the smallest overlapping scores. A robust version of *Apiletti et al. (2012)* can be found in *Mahmoud et al. (2014)*, where expression interval for each gene is calculated by using the Interquartile Range. *Mahmoud et al. (2014)* also considered the proportion of overlapping samples (POS) in each class for each gene. Genes with lower POS, i.e., proportional overlapping scores were considered informative. After obtaining the POS, the relative dominant class (RDC) for each gene was also calculated which associates each gene with the class for which it has a stronger distinguishing capability. The final set of genes/features is obtained by combining the minimum gene set via gene masks top ranked genes based on proportional overlapping scores (POS). *Li & Gu (2015)* proposed a

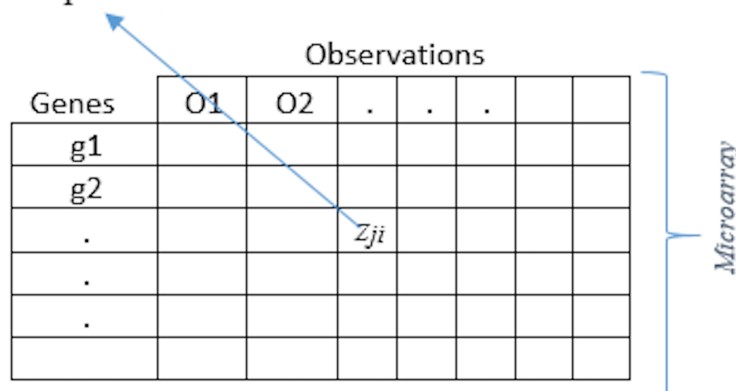

**Figure 1 Gene expression data.**

method called more relevance less redundancy algorithm. Another study by *Nardone, Ciaramella & Staiano (2019)* introduced a two step procedure for the feature selection, where extensive experiments were performed to evaluate the performance of their proposed method on the publicly available datasets related to computational biology field. A novel supervised learning technique is introduced in *Bidgoli, Ebrahimpour-Komleh & Rahnamayan (2020)*. This method is designed particularly for the multi class problems. Furthermore this method is an extended version of decomposition-based multi-objective optimization approach. A feature selection method for binary classification problems was introduced by *Dashtban, Balafar & Suravajhala (2018)*, in which the traditional bat algorithm is extended with more refined formulations, improved and multi-objective operators and a novel local search strategy. Other examples of feature selection methods could be found in *MotieGhader et al. (2020)*, *Dashtban & Balafar (2017)*, *Nematzadeh et al. (2019)*, *Maghsoudloo et al. (2020)*, *Rostami et al. (2020)*, *Shamsara & Shamsara (2020)*, *Ao et al. (2020)*, *Statnikov et al. (2005)*, *Rana et al. (2019)*, *Chamikara et al. (2016)*, *Nardone, Ciaramella & Staiano (2019)* and the references cited therein.

## METHOD

Microarray gene expression data is usually in the form of a matrix, i.e., $Z = [z_{ji}]$, where $Z \in \mathbb{R}^{p \times n}$ and $z_{ji}$ is the observed expression value of $j^{th}$ gene for $i^{th}$ tissue sample, for $j = 1,2,3,\ldots,p$ and $i = 1,2,3,\ldots,n$. Each tissue sample is categorized into one of the two classes, i.e., 0 or 1. Let $W \in \mathbb{R}^n$ be the class labels vector such that its $i^{th}$ component $w_i$ takes a unique value $c$ which is in the form of either 0 or 1. The number of samples/observations in microarray gene expression datasets are usually smaller than the number of features, which is also called $n < p$ problem. Figure 1 represents the common layout of a gene expression dataset. Observations/samples are listed in the rows while the genes are given in the columns. Corresponding to each sample the gene expression values for each gene are given in the cells.

Further definitions used in this paper are given below:

## Class interval

For each class and gene $j$, two expression intervals are defined as;

$$R_{j,c} = [d_{j,c}, e_{j,c}], j = 1, 2, 3, \ldots, p, c = 0, 1. \tag{1}$$

such that $d_{j,c} = Q_{1(j,c)} - 0.9MAD_{(j,c)}$ and $e_{j,c} = Q_{3(j,c)} + 0.9MAD_{(j,c)}$ where $Q_{1(j,c)}$, $Q_{3(j,c)}$ and $MAD_{(j,c)}$ are the first (lower) quartile, third (upper) quartile and median absolute deviation (MAD) of gene $j$ for class $c$ respectively.

## Overlapped region

The overlapping region between the two classes is represented by $R_j^v$, which shows the intersection region between the expression values of the target classes for gene $j$. It is defined by;

$$R_j^v = R_{j,1} \cap R_{j,2}. \tag{2}$$

## Non-outlier sample set

The non-outlier sample set is symbolized by $N_j$, it is a set of observations with expression values lying within their own response class core intervals. It is given as:

$$N_j = i : z_{ji} \in R_{j,c_i}, i = 1, 2, 3, \ldots, n. \tag{3}$$

## Total core interval

Total core interval for gene $j$ is denoted by $R_j$, it is the area between a global minimum and global maximum boundaries of both classes' core intervals. It is given as:

$$R_j = [d_j, e_j], \tag{4}$$

such that $d_j = \min(d_{j,1}, d_{j,2})$, $e_j = max(e_{j,1}, e_{j,2})$ represent lowest and highest boundaries of core interval $R_{j,c}$ of gene/feature with response $c = (0,1)$ respectively.

## *Non-overlapped sample set*

For gene $j$, the non-overlapping set is represented by $O'_j$, which contains the non-outlier samples given by $N_j$, with expression values not falling inside the overlap interval. It is given as:

$$O'_j = \{i : i \in N_j \wedge z_{ji} \in R_{j,1} \ominus R_{j,2}\}. \tag{5}$$

## *Overlapped sample set*

The overlapping samples set for gene $j$ is characterized by $O_j$, which consists of the observations with expression values falling inside the overlap interval $R^v_j$. It is given as:

$$O_j = N_j - O'_j, \tag{6}$$

where $O'_j$ contains all the non-overlapping samples.

### Gene masks matrix

The matrix of gene masks, i.e., $M = [m_{ji}]_{p \times n}$ is constructed as follows:

$$m_{ji} = \begin{cases} 1, & if \ z_{ji} \in R_{j,1} \cap R_{j,2}, \\ 0, & otherwise, j = 1, 2, 3, \ldots, p, \end{cases} \tag{7}$$

such that $R_{j,1} = [d_{j,1}, e_{j,1}]$ and $R_{j,2} = [d_{j,2}, e_{j,2}]$, $d_{j,1} = Q_{1(j,1)} - 0.9 MAD_{(j,1)}$, $e_{j,1} = Q_{3(j,1)} + 0.9 MAD_{(j,1)}$, $d_{j,2} = Q_{1(j,2)} - 0.9 MAD_{(j,2)}$ and $e_{j,2} = Q_{3(j,2)} + 0.9 MAD_{(j,2)}$ respectively.

In the above expressions $Q_{1(j,c)}$, $Q_{3(j,c)}$ and $MAD_{(j,c)}$ represent the lower (first) quartile, upper (third) quartile and median absolute deviation respectively for each class $c$, where $c$ is either 0 or 1.

### Relative dominant class (RDC)

For each gene, Relative dominant class (RDC) is calculated, which associates each feature/ gene with the class it is more capable to differentiate. It is defined as:

$$RDC_j = argmax_c \left[ \frac{\sum_{j \in U_c} I(m_{ji} = 1)}{|U_c|} \right], \tag{8}$$

where $U_c$ represents class $c$ samples set, i.e., $U_c \in \{i \ c_j = c\}$.

### Proposed (RPOS) score

The proposed method (RPOS) is defined as.

$$RPOS_j = 4 \frac{|O_j|}{|N_j|} \left( \prod_{c=1}^{2} \phi_c \right), \tag{9}$$

where $< R_j^v >$ is the length of overlap interval, $< R_j >$ is the length of total core interval, $|O_j|$ is the total number of overlapped samples and $|N_j|$ is the total number of non-outlier samples for gene $j$. $\phi_c = \frac{|O_{j,c}|}{|O_j|}$, where $|O_{j,c}|$ represents the overlapped samples lying in class. The number 4 is multiplied to keep the RPOS scores between 0 and 1. Smaller value of RPOS represents that a particular gene is more informative in classifying the tissue sample to its correct class.

The proposed method thus takes the following steps in selecting the most discriminative genes.

1. The proposed method initially identifies the minimum subset of genes via the greedy approach given in *Apiletti et al. (2012)*. The greedy approach utilizes gene mask matrix given in Eq. (7) and RPOS scores in Eq. (9) to form this subset. Gene that has the highest number of bits equals 1 is included in the subset. If more than one genes having the same number of bits 1 exist the one with smaller RPOS is selected. Using AND operator, the gene masks of the remaining genes are updated for the selection of the second gene and so on. This process is repeated until the desired number of genes are selected, or the genes have no 1's in their gene masks. For further details on greedy approach gene selection, see *Apiletti et al. (2012)*.

2. The genes that are not selected in the minimum subset are arranged according to the RPOS scores and relative dominant class (RDC) by round-robin fashion method in ascending order. A smaller score represents the higher discriminative ability of gene/feature.

3. After arranging the genes in step (2), the required top most ranked genes are selected.

4. The final set of genes for the model construction is obtained by combining the genes in steps (1) and (3).

The general workflow of the proposed (RPOS) method, along with its pseudo-code, is given in Fig. 2 and Algorithm 1, respectively.

The proposed (RPOS) method is novel in the sense that it utilizes median absolute deviation (MAD) for the construction of core expression intervals of the expression values of genes. The drawback of POS in Mahmoud et al. (2014) is that the gene masks are calculated on the basis of expression intervals by using the interquartile range approach. The construction of gene masks can be affected by outliers because of the smaller breakdown point, i.e., 25% of interquartile range. The breakdown point of (MAD) is 50%, which is less vulnerable to outliers, thereby reducing the effect of outliers while constructing the gene masks.

# EXPERIMENTS AND RESULTS

This section provides a detailed description of the experiments executed for assessing the proposed method against the other methods on benchmark gene expression datasets. A common practice for investigating the efficacy of gene selection methods is to check the discriminative ability of the selected genes by using different classifiers. This is usually done by recording classification accuracy of the classifiers applied on datasets with selected genes only while discarding the rest of the genes. Golub et al. (1999) have used different feature/gene selection techniques given in Statnikov et al. (2005), and it has been observed that gene selection methods have a significant effect on the classifier's accuracy. This approach has been widely used in several other studies (Apiletti et al., 2012; Mahmoud et al., 2014). Before listing the results from the analyses done in this paper following the above-mentioned approach, a brief description of datasets is given below.

## Microarray gene expression datasets

In this research work, a total of 10 microarray gene expression datasets are taken as standard benchmark binary classification problems. These datasets are taken from various open sources with a varying number of genes and observations. A brief description of the benchmark datasets used in the current paper is given in Table 1. The table provides the number of samples, number of genes, class-wise distribution of samples in the data and source against each dataset.

## Experimental setup

Experimental setup for the analyses done in the paper is as follow. The datasets considered in this study are divided into two mutually exclusive parts in the following manner: In the

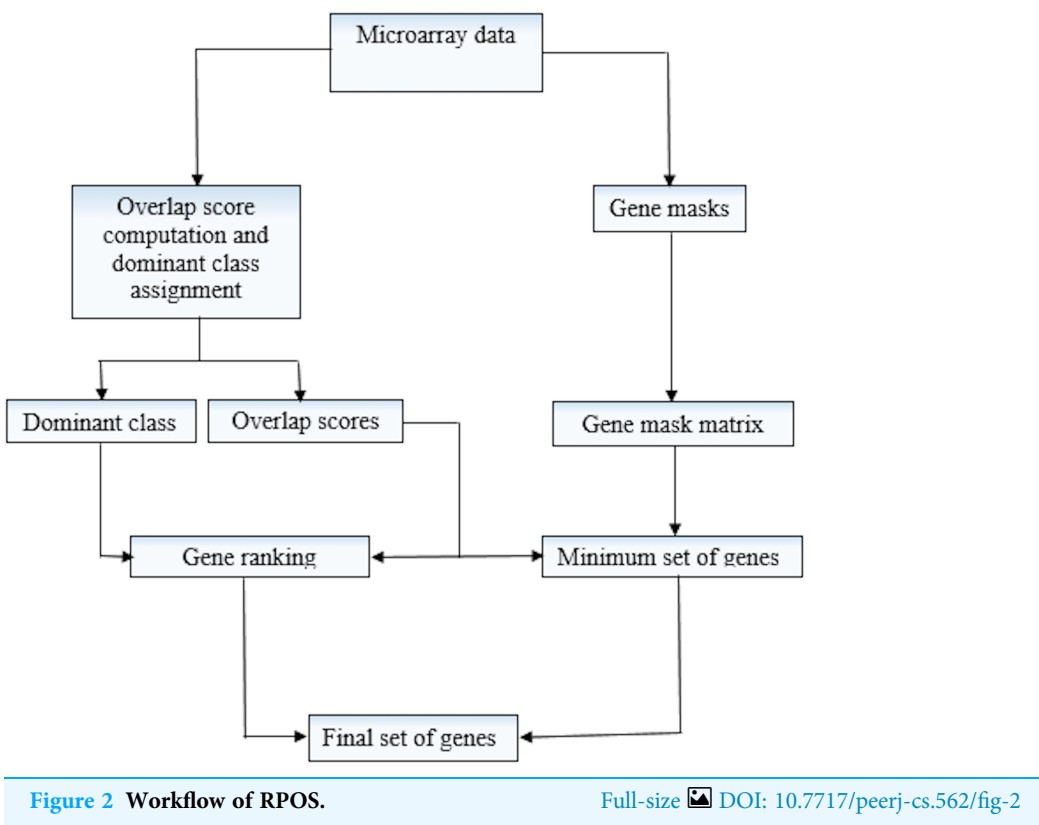

**Figure 2 Workflow of RPOS.**               

first part, seventy percent (70%) of the observations from each dataset randomly selected without replacement are considered as training part, while the remaining thirty percent (30%) of the observations are considered as a testing part. In the second part, thirty percent (30%) of the observations in each dataset randomly selected without replacement are considered as training part. In comparison, the remaining seventy percent (70%) of the observation are considered as testing part. A split sample analysis of 500 runs is carried out for each combination of gene selection methods and the corresponding classifiers using 70% training, 30% testing and 30% training, 70% testing partitions. The classifiers which are considered in this study are Random forest (RF), support vector machine (SVM) and k-Nearest neighbours (k-NN). For Random forest, R package, i.e., `randomForest` (*Liaw & Wiener, 2002*) is used with default parameters `ntree` = 500, `mtry` = $\sqrt{p}$ and `nodesize` = 1. For the implementation of support vector machine R package `kernlab` (*Karatzoglou et al., 2004*) is used with default parameters. Similarly for k-Nearest neighbor classifier R package `caret` from *Jed Wing et al. (2019)* is used the default parameter value of $k = 5$. Using the training parts of each dataset, a set of discriminative genes, i.e., 5, 10, 15, 20, 25 and 30 are selected by different gene selection methods to train the classifiers. Gene selection methods considered in this paper are Wilcoxon Rank Sum Test (*Liao, Li & Luo, 2006*), Proportional Overlapping Score (POS) based method (*Mahmoud et al., 2014*), Genes Selection by Clustering (GClust) (*Khan et al., 2019*), Maximum Relevance Minimum Redundancy (mRmR) (*Ding & Peng, 2005*) and Significant Features by SVM and t-test (sigF) (*Das et al., 2020*). The performance of the selected genes are

---

**Algorithm 1** Algorithm of RPOS Method For Gene Selection.

1: **Inputs:** $X, Y$ and number genes ($r$) to be selected.

2: **Output:** Sequence of selected genes $T$.

3: **for all** $j \in$ H **do**

4:     **for** $c = 0,1$ **do**

5:         Compute the relative dominant class for each gene, i.e., $R(j,c)$ in Eq. (1).

6:     **end for**

7:     **for** $i \rightarrow N$ **do**

8:         Compute the gene mask for each gene, i.e., $m_{ji}$ as defined in Eq. (7).

9:         Compute the $RPOS_j$ scores for each gene as defined in Eq. (9).

10:       Assign $RDC_J$ to each gene as defined in Eq. (8).

11:     **end for**

12:     let $M \in \mathbb{R}^{P \times N}$ be the gene mask matrix $M = [m_{ji}]$, where its $i^{th}$ value for $j^{th}$ gene is either 0 or 1.

13:     Compute the total or aggregate mask of genes and denote it by $M..(H)$.

14:     Use the Greedy search approach to select the minimum subset of genes from $M$, $M..(H)$ and $RPOSj$ and denote it by $H^*$.

15:     Perform $H = H - H^*$, this will exclude the genes selected in minimum subset from the whole set of genes.

16:     Arrange the genes in $RDC_j$ in the increasing order of $RPOS_J$ for each class.

17: **end for**

18: **Obtaining final listed or ranked genes.**

19: **if** $r \leq |H^*|$ **then**

20:    Then $T$ includes the genes which are first $r$ genes in $H^*$.

21:    **while** $|T| < r$ **do**

22:       Increase $T$ by one gene in a round-robin fashion method.

23:    **end while**

24: **end if**

25: **return** $T$

---

investigated by the average values of the performance metrics, i.e., classification error rate, Brier score and sensitivity using the testing parts of each dataset.

## RESULTS AND DISCUSSION

The results of the proposed method and other methods included in this study are obtained for all the datasets. The results of three datasets, i.e., "TumorC", "Breast" and "Srbct" are given in Tables 2, 3 and 4. These results are based on 70% training and 30% testing parts portioning of the datasets. The results of the remaining eight datasets are given in Supplemental File (Tables S1–S15). From Table 2 given below, it is clear that for "TumorC" dataset the proposed method (RPOS) performed better than all the other methods in terms of all the performance metrics considered, except the Wilcoxon rank-sum test, which performed better for the number of genes, i.e., 5, 10, 15 and 20 on Support vector machine

**Table 1 Datasets description showing number of samples, number of genes, class wise distribution of samples in the data.**

| Dataset | Samples | Genes | Class wise distribution | Source |
|---|---|---|---|---|
| Leukeamia | 68 | 7,029 | 49/23 | *Alon et al. (1999)* |
| nki | 144 | 76 | 96/48 | *Karatzoglou et al. (2004)* |
| Colon | 62 | 2,000 | 40/22 | *Golub et al. (1999)* |
| Breast | 78 | 4,948 | 34/44 | *Michiels, Koscielny & Hill (2005)* |
| GSE4045 | 37 | 22,215 | 29/8 | *Laiho et al. (2007)* |
| Prostate | 412 | 10,936 | 343/69 | *Statnikov et al. (2005)* |
| Srbct | 54 | 2,308 | 28/25 | *Statnikov et al. (2005)* |
| Lung | 148 | 12,600 | 134/14 | *Gordon et al. (2002)* |
| DLBCL | 76 | 7,070 | 58/19 | https://file.biolab.si/biolab/supp/bi-cancer/projections/info/DLBCL.html |
| TumorC | 60 | 7,129 | 39/21 | https://www.openml.org |

**Table 2 Classification error rate, sensitivity and Brier score produced by Random Forest, k-Nearest Neighbors and Support Vector Machine classifiers on TumorC dataset based on genes selected by the given methods. The best result is shown in bold.**

| Genes | | RF | | | | | | kNN | | | | | | SVM | | | | | |
|---|---|---|---|---|---|---|---|---|---|---|---|---|---|---|---|---|---|---|---|
| | | POS | RPOS | GClust | sigF | Wilc | mRmR | POS | RPOS | GClust | sigF | Wilc | mRmR | POS | RPOS | GClust | sigF | Wilc | mRmR |
| 5 | Err | 0.362 | **0.221** | 0.334 | 0.482 | 0.450 | 0.451 | 0.423 | **0.269** | 0.383 | 0.407 | 0.398 | 0.396 | 0.362 | **0.264** | 0.333 | 0.277 | 0.442 | 0.373 |
| | BS | 0.013 | **0.011** | 0.023 | 0.274 | 0.268 | 0.276 | **0.017** | **0.017** | 0.026 | 0.259 | 0.260 | 0.257 | 0.035 | **0.012** | 0.037 | 0.188 | 0.254 | 0.244 |
| | sen | 0.311 | **0.643** | 0.363 | 0.217 | 0.236 | 0.261 | 0.344 | **0.630** | 0.454 | 0.346 | 0.348 | 0.389 | 0.557 | 0.700 | 0.579 | **0.773** | 0.070 | 0.189 |
| 10 | Err | 0.336 | **0.257** | 0.313 | 0.482 | 0.401 | 0.348 | 0.332 | **0.220** | 0.355 | 0.471 | 0.391 | 0.395 | 0.341 | **0.242** | 0.336 | 0.302 | 0.396 | 0.349 |
| | BS | **0.015** | **0.015** | 0.022 | 0.274 | 0.252 | 0.231 | 0.019 | **0.015** | 0.029 | 0.282 | 0.254 | 0.260 | 0.086 | **0.015** | 0.072 | 0.204 | 0.241 | 0.230 |
| | sen | 0.358 | **0.569** | 0.375 | 0.217 | 0.278 | 0.408 | 0.427 | **0.722** | 0.505 | 0.332 | 0.365 | 0.387 | 0.532 | 0.699 | 0.589 | **0.790** | 0.150 | 0.268 |
| 15 | Err | 0.351 | **0.288** | 0.312 | 0.482 | 0.391 | 0.338 | 0.293 | **0.242** | 0.344 | 0.415 | 0.382 | 0.386 | 0.312 | 0.249 | 0.311 | **0.228** | 0.400 | 0.351 |
| | BS | 0.016 | **0.014** | 0.026 | 0.274 | 0.250 | 0.228 | 0.018 | **0.014** | 0.039 | 0.272 | 0.249 | 0.249 | 0.052 | **0.013** | 0.066 | 0.166 | 0.245 | 0.232 |
| | sen | 0.286 | **0.523** | 0.399 | 0.217 | 0.177 | 0.439 | 0.462 | **0.652** | 0.511 | 0.246 | 0.363 | 0.398 | 0.472 | 0.714 | 0.588 | **0.824** | 0.095 | 0.279 |
| 20 | Err | 0.297 | **0.274** | 0.303 | 0.482 | 0.464 | 0.371 | 0.305 | **0.272** | 0.345 | 0.426 | 0.393 | 0.383 | 0.270 | **0.208** | 0.313 | 0.233 | 0.387 | 0.387 |
| | BS | **0.015** | **0.015** | 0.021 | 0.274 | 0.266 | 0.234 | **0.016** | 0.017 | 0.033 | 0.280 | 0.260 | 0.255 | 0.074 | **0.014** | 0.054 | 0.170 | 0.249 | 0.247 |
| | sen | 0.440 | **0.553** | 0.404 | 0.217 | 0.091 | 0.312 | 0.478 | **0.620** | 0.555 | 0.245 | 0.347 | 0.365 | 0.561 | 0.721 | 0.666 | **0.754** | 0.033 | 0.115 |
| 25 | Err | 0.306 | **0.286** | 0.300 | 0.482 | 0.423 | 0.373 | 0.336 | **0.281** | 0.333 | 0.459 | 0.377 | 0.392 | 0.281 | 0.217 | 0.296 | **0.211** | 0.379 | 0.378 |
| | BS | 0.015 | **0.013** | 0.026 | 0.274 | 0.263 | 0.237 | 0.018 | **0.015** | 0.028 | 0.284 | 0.245 | 0.253 | 0.031 | **0.012** | 0.049 | 0.157 | 0.245 | 0.250 |
| | sen | 0.399 | **0.497** | 0.411 | 0.217 | 0.120 | 0.257 | 0.364 | **0.623** | 0.539 | 0.262 | 0.331 | 0.348 | 0.518 | 0.716 | 0.678 | **0.821** | 0.044 | 0.062 |
| 30 | Err | 0.335 | **0.275** | 0.309 | 0.482 | 0.441 | 0.379 | 0.373 | **0.302** | 0.331 | 0.467 | 0.388 | 0.400 | 0.283 | **0.226** | 0.286 | 0.213 | 0.380 | 0.384 |
| | BS | 0.014 | **0.012** | 0.020 | 0.274 | 0.263 | 0.240 | 0.020 | **0.016** | 0.025 | 0.282 | 0.253 | 0.259 | 0.024 | **0.014** | 0.039 | 0.151 | 0.252 | 0.248 |
| | sen | 0.317 | **0.505** | 0.423 | 0.217 | 0.064 | 0.304 | 0.304 | **0.573** | 0.560 | 0.174 | 0.331 | 0.382 | 0.480 | 0.665 | 0.661 | **0.854** | 0.019 | 0.066 |

classifier in terms of classification error rate. Similarly, from Table 3, it is evident that for "Breast" dataset the proposed method (RPOS) outperformed the other methods on all the classifiers. Table 4, gives the results for the dataset "Srbct", where the proposed method (RPOS) shows better results for the number of genes genes, i.e., 5 on on Random forest (RF) classifier than the other methods. For the number of genes 10, the Wilcoxon rank-sum test performs better in terms of the classification error rate. In contrast, in terms of

**Table 3** Classification error rate, sensitivity and Brier score produced by Random Forest, k-Nearest Neighbors and Support Vector Machine classifiers on Breastcancer dataset based on genes selected by the given methods. The best result is shown in bold.

| Genes | | RF | | | | | | KNN | | | | | | SVM | | | | | |
|---|---|---|---|---|---|---|---|---|---|---|---|---|---|---|---|---|---|---|---|
| | | POS | RPOS | GClust | sigF | Wilc | mRmR | POS | RPOS | GClust | sigF | Wilc | mRmR | POS | RPOS | GClust | sigF | Wilc | mRmR |
| 5 | Err | 0.296 | **0.239** | 0.261 | 0.490 | 0.390 | 0.455 | 0.314 | **0.206** | 0.313 | 0.448 | 0.405 | 0.402 | 0.310 | **0.260** | 0.512 | 0.522 | 0.384 | 0.367 |
| | BS | 0.013 | **0.010** | 0.165 | 0.287 | 0.254 | 0.277 | 0.014 | **0.011** | 0.290 | 0.275 | 0.261 | 0.254 | 0.021 | **0.011** | 0.262 | 0.251 | 0.262 | 0.244 |
| | sen | 0.784 | **0.837** | 0.810 | 0.621 | 0.722 | 0.661 | 0.706 | **0.862** | 0.798 | 0.703 | 0.761 | 0.760 | 0.704 | **0.776** | 0.558 | 0.506 | 0.714 | 0.791 |
| 10 | Err | 0.308 | **0.224** | 0.261 | 0.514 | 0.360 | 0.462 | 0.276 | **0.240** | 0.297 | 0.501 | 0.390 | 0.396 | 0.272 | **0.214** | 0.522 | 0.484 | 0.351 | 0.456 |
| | BS | 0.013 | **0.011** | 0.168 | 0.278 | 0.225 | 0.266 | **0.013** | **0.013** | 0.202 | 0.304 | 0.251 | 0.254 | 0.022 | **0.013** | 0.261 | 0.260 | 0.237 | 0.260 |
| | sen | 0.757 | **0.858** | 0.818 | 0.613 | 0.709 | 0.654 | 0.786 | **0.842** | 0.819 | 0.677 | 0.754 | 0.764 | 0.750 | **0.796** | 0.575 | 0.412 | 0.704 | 0.743 |
| 15 | Err | 0.323 | **0.179** | 0.202 | 0.519 | 0.337 | 0.414 | 0.297 | **0.204** | 0.241 | 0.514 | 0.391 | 0.401 | 0.262 | **0.215** | 0.515 | 0.462 | 0.350 | 0.427 |
| | BS | 0.013 | **0.009** | 0.145 | 0.275 | 0.222 | 0.246 | 0.012 | **0.008** | 0.182 | 0.324 | 0.255 | 0.256 | 0.046 | **0.010** | 0.262 | 0.260 | 0.235 | 0.255 |
| | sen | 0.709 | **0.864** | 0.848 | 0.643 | 0.767 | 0.719 | 0.781 | **0.835** | 0.810 | 0.685 | 0.768 | 0.763 | 0.741 | **0.781** | 0.564 | 0.354 | 0.742 | 0.798 |
| 20 | Err | 0.290 | **0.195** | 0.199 | 0.481 | 0.377 | 0.468 | 0.279 | **0.207** | 0.257 | 0.473 | 0.408 | 0.395 | 0.225 | **0.215** | 0.542 | 0.409 | 0.386 | 0.474 |
| | BS | 0.014 | **0.011** | 0.155 | 0.265 | 0.234 | 0.258 | 0.014 | **0.010** | 0.188 | 0.284 | 0.260 | 0.256 | 0.041 | **0.013** | 0.259 | 0.254 | 0.251 | 0.265 |
| | sen | 0.767 | **0.853** | 0.851 | 0.694 | 0.717 | 0.686 | 0.794 | 0.815 | **0.840** | 0.734 | 0.745 | 0.763 | 0.793 | **0.798** | 0.526 | 0.390 | 0.673 | 0.781 |
| 25 | Err | 0.300 | **0.186** | 0.223 | 0.495 | 0.366 | 0.473 | 0.256 | **0.186** | 0.271 | 0.462 | 0.404 | 0.393 | 0.250 | 0.223 | 0.523 | 0.406 | 0.377 | 0.427 |
| | BS | 0.012 | **0.010** | 0.156 | 0.270 | 0.229 | 0.265 | 0.012 | **0.010** | 0.178 | 0.279 | 0.260 | 0.251 | 0.033 | **0.010** | 0.264 | 0.254 | 0.246 | 0.259 |
| | sen | 0.777 | **0.883** | 0.832 | 0.694 | 0.726 | 0.659 | 0.801 | 0.829 | **0.838** | 0.693 | 0.753 | 0.759 | 0.790 | **0.798** | 0.567 | 0.397 | 0.691 | 0.790 |
| 30 | Err | 0.268 | **0.197** | 0.242 | 0.411 | 0.350 | 0.454 | 0.261 | **0.192** | 0.258 | 0.436 | 0.387 | 0.394 | 0.249 | **0.198** | 0.455 | 0.418 | 0.351 | 0.457 |
| | BS | 0.009 | **0.008** | 0.158 | 0.248 | 0.222 | 0.261 | 0.011 | **0.009** | 0.182 | 0.281 | 0.250 | 0.253 | 0.031 | **0.009** | 0.260 | 0.253 | 0.236 | 0.262 |
| | sen | 0.823 | **0.870** | 0.836 | 0.733 | 0.736 | 0.694 | 0.813 | **0.834** | 0.818 | 0.708 | 0.776 | 0.767 | 0.787 | **0.794** | 0.661 | 0.365 | 0.698 | 0.767 |

**Table 4** Classification error rate, sensitivity and Brier score produced by Random Forest, k-Nearest Neighbors and Support Vector Machine classifiers on srbct dataset based on genes selected by the given methods. The best result is shown in bold.

| Genes | | RF | | | | | | kNN | | | | | | SVM | | | | | |
|---|---|---|---|---|---|---|---|---|---|---|---|---|---|---|---|---|---|---|---|
| | | POS | RPOS | GClust | sigF | Wilc | mRmR | POS | RPOS | GClust | sigF | Wilc | mRmR | POS | RPOS | GClust | sigF | Wilc | mRmR |
| 5 | Err | 0.048 | **0.019** | 0.096 | 0.040 | 0.021 | 0.390 | 0.078 | 0.034 | 0.100 | **0.000** | 0.074 | 0.078 | 0.086 | 0.021 | 0.035 | **0.007** | 0.328 | 0.412 |
| | BS | 0.005 | **0.002** | 0.096 | 0.029 | 0.023 | 0.236 | 0.007 | **0.001** | 0.057 | 0.002 | 0.071 | 0.074 | 0.037 | **0.003** | 0.028 | 0.011 | 0.217 | 0.255 |
| | sen | 0.919 | **0.988** | 0.961 | 0.980 | 0.978 | 0.549 | **1.000** | **1.000** | 0.718 | **1.000** | 0.915 | 0.914 | 0.878 | **0.998** | 0.942 | 0.984 | 0.608 | 0.574 |
| 10 | Err | 0.018 | 0.021 | 0.027 | 0.035 | **0.013** | 0.086 | 0.039 | 0.038 | 0.055 | **0.000** | 0.071 | 0.069 | 0.016 | 0.011 | 0.029 | **0.006** | 0.204 | 0.143 |
| | BS | **0.002** | 0.003 | 0.029 | 0.027 | 0.022 | 0.089 | 0.004 | 0.002 | 0.041 | **0.000** | 0.076 | 0.071 | 0.016 | **0.002** | 0.031 | 0.013 | 0.138 | 0.093 |
| | sen | **0.999** | 0.991 | 0.957 | 0.981 | 0.977 | 0.879 | **1.000** | **1.000** | 0.852 | **1.000** | 0.925 | 0.918 | 0.992 | 0.995 | 0.943 | **0.998** | 0.766 | 0.785 |
| 15 | Err | 0.004 | 0.014 | 0.016 | **0.001** | 0.013 | 0.165 | 0.039 | 0.035 | 0.075 | **0.000** | 0.074 | 0.071 | 0.004 | 0.004 | 0.015 | **0.002** | 0.188 | 0.182 |
| | BS | **0.002** | **0.002** | 0.028 | 0.021 | 0.024 | 0.142 | 0.002 | 0.002 | 0.047 | **0.000** | 0.071 | 0.073 | 0.005 | **0.001** | 0.015 | 0.010 | 0.118 | 0.129 |
| | sen | 0.995 | 0.991 | 0.956 | **0.998** | 0.977 | 0.805 | **1.000** | **1.000** | 0.807 | **1.000** | 0.927 | 0.910 | 0.995 | **1.000** | 0.962 | **1.000** | 0.756 | 0.764 |
| 20 | Err | 0.009 | **0.007** | 0.016 | 0.010 | 0.009 | 0.081 | 0.036 | 0.036 | 0.053 | **0.000** | 0.066 | 0.071 | 0.011 | 0.003 | 0.020 | **0.002** | 0.144 | 0.130 |
| | BS | **0.002** | **0.002** | 0.029 | 0.021 | 0.023 | 0.088 | 0.002 | 0.002 | 0.041 | **0.000** | 0.069 | 0.072 | 0.007 | **0.001** | 0.019 | 0.010 | 0.098 | 0.082 |
| | sen | 0.987 | 0.990 | 0.956 | **1.000** | 0.986 | 0.875 | **1.000** | **1.000** | 0.895 | **1.000** | 0.919 | 0.911 | 0.997 | 0.999 | 0.986 | **1.000** | 0.797 | 0.816 |
| 25 | Err | 0.009 | **0.004** | 0.017 | 0.011 | 0.009 | 0.067 | 0.038 | 0.020 | 0.060 | **0.000** | 0.066 | 0.074 | 0.011 | 0.008 | 0.030 | **0.000** | 0.134 | 0.098 |
| | BS | **0.002** | **0.002** | 0.031 | 0.021 | 0.024 | 0.084 | 0.002 | **0.001** | 0.039 | 0.001 | 0.071 | 0.072 | 0.006 | **0.002** | 0.023 | 0.008 | 0.087 | 0.067 |
| | sen | 0.992 | 0.997 | 0.956 | **1.000** | 0.987 | 0.881 | **1.000** | **1.000** | 0.870 | **1.000** | 0.923 | 0.915 | 0.999 | 0.997 | 0.977 | **1.000** | 0.826 | 0.885 |
| 30 | Err | **0.006** | **0.006** | 0.023 | 0.007 | 0.005 | 0.075 | 0.034 | 0.002 | 0.047 | **0.000** | 0.064 | 0.065 | 0.009 | 0.014 | 0.018 | **0.000** | 0.131 | 0.129 |
| | BS | **0.002** | **0.002** | 0.029 | 0.022 | 0.024 | 0.094 | 0.002 | **0.001** | 0.040 | 0.001 | 0.069 | 0.070 | 0.006 | **0.002** | 0.017 | 0.006 | 0.087 | 0.090 |
| | sen | 0.992 | 0.997 | 0.957 | **1.000** | 0.994 | 0.883 | **1.000** | **1.000** | 0.866 | **1.000** | 0.914 | 0.924 | 0.998 | 0.999 | 0.951 | **1.000** | 0.828 | 0.855 |

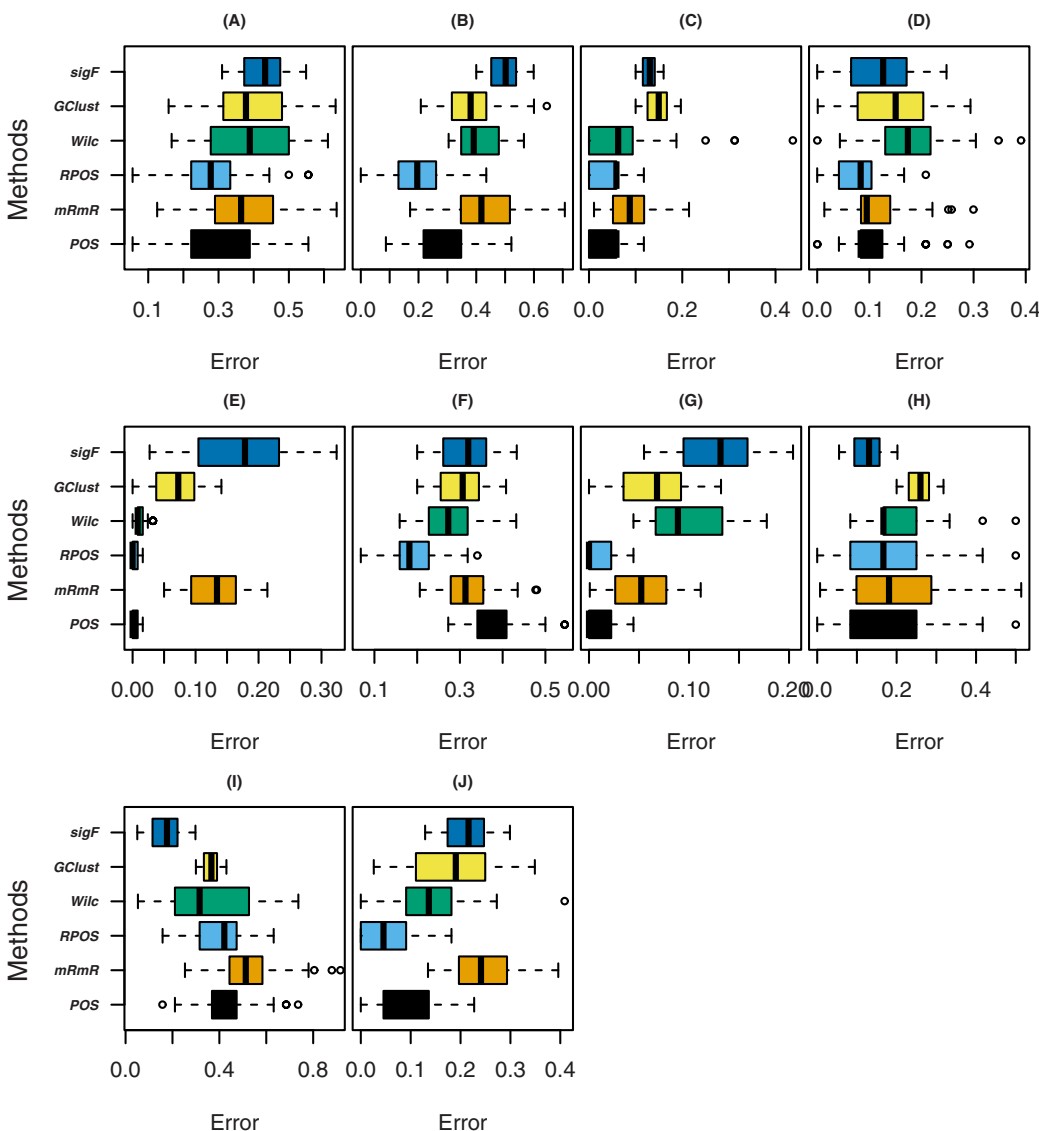

**Figure 3 Boxplots of classification error rates for 20 number of genes for the datasets; (A) TumorC, (B) Breast, (C) srbct, (D) DLBCL, (E) Prostate, (F) nki, (G) Lung, (H) GSE4045, (I) Colon and (J) Leukaemia.**

Brier score and sensitivity, the POS method performs better than all the other methods. For a set of 15 discriminative genes, POS outperforms all the other methods on Random forest classifier. For the rest of the gene numbers, i.e., 20, 25 and 30, the proposed method outperforms all the other methods on Random forest classifier. In the case of k-Nearest neighbours classifier, the proposed method RPOS gives similar results in terms of sensitivity for the number of genes 5 and 10. Similarly, for the number of genes 15, the results of the proposed method RPOS and POS are the same. For a set of 20 discriminative genes, the performance of the proposed method RPOS and POS equally performs in terms of classification error rate, Brier score and sensitivity. The proposed method RPOS outperforms all the other methods for the set of genes, i.e., 25 and 30. On support vector

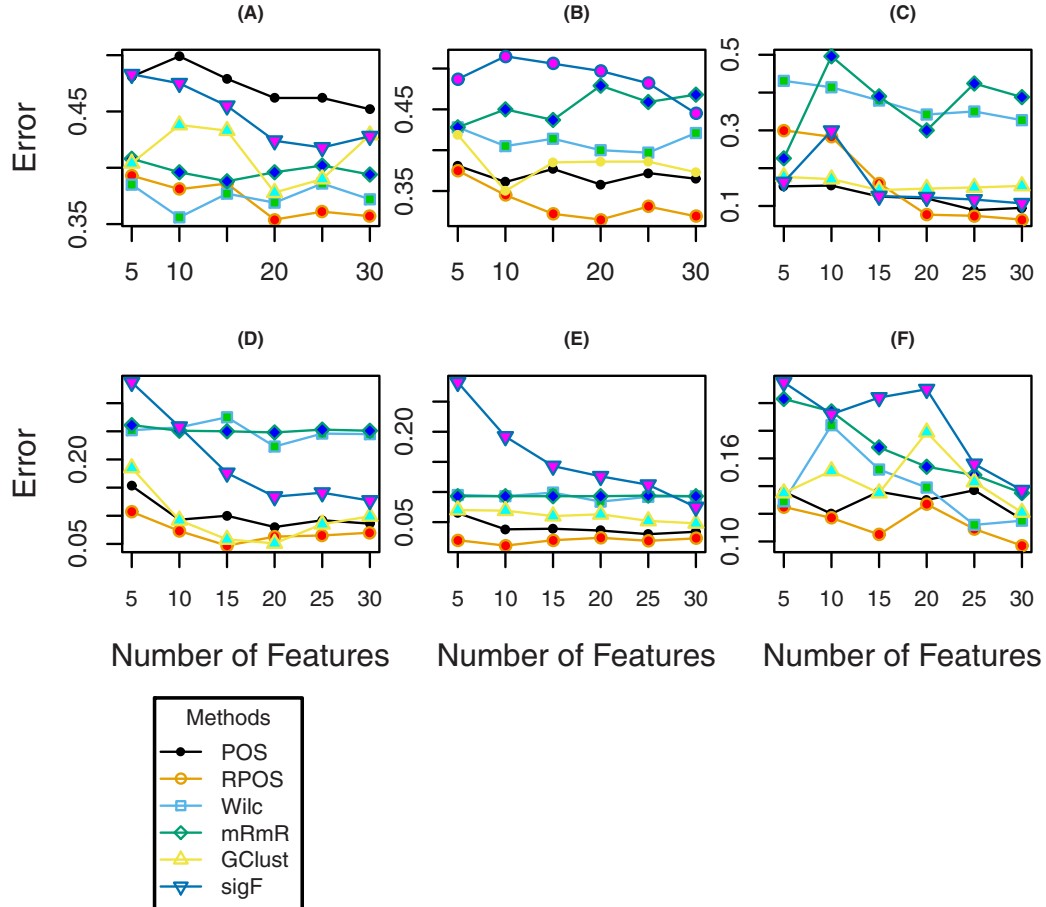

**Figure 4 Classification error rates of the methods for different number of genes for the datasets; (A) TumorC, (B) Breast, (C) srbct, (D) DLBCL, (E) Lung and (F) Leukaemia.**

machine (SVM) classifier, the proposed method RPOS outperforms all the other methods except for the set of genes 25 and 30 where the method POS performs better than all the other methods in terms of sensitivity and classification error rate.

Boxplots of the results of the proposed method and the other methods for twenty number of genes are also constructed given in Fig. 3. From the boxplots in Fig. 3, it is clear that the proposed method (RPOS) outperforms all the other methods except for the datasets "Srbct" and "Prostate" where the proposed method RPOS and the method POS almost provide similar results. In the case of the dataset "GSE4045" the sigF method outperforms all the other methods. Similarly, in the case of dataset "Colon", the performance of the proposed method RPOS and the method POS is similar, while the method sigF outperforms all the other methods. The proposed method RPOS outperforms the rest of the methods on the dataset "Leukaemia". Overall the proposed method RPOS outperforms all the other methods on 5 out of 10 datasets and provides similar results to that of the method POS on 3 datasets.

To further investigate the efficiency of the proposed method RPOS, and the other methods, plots of classification error rates, Brier Scores and sensitivity for a various

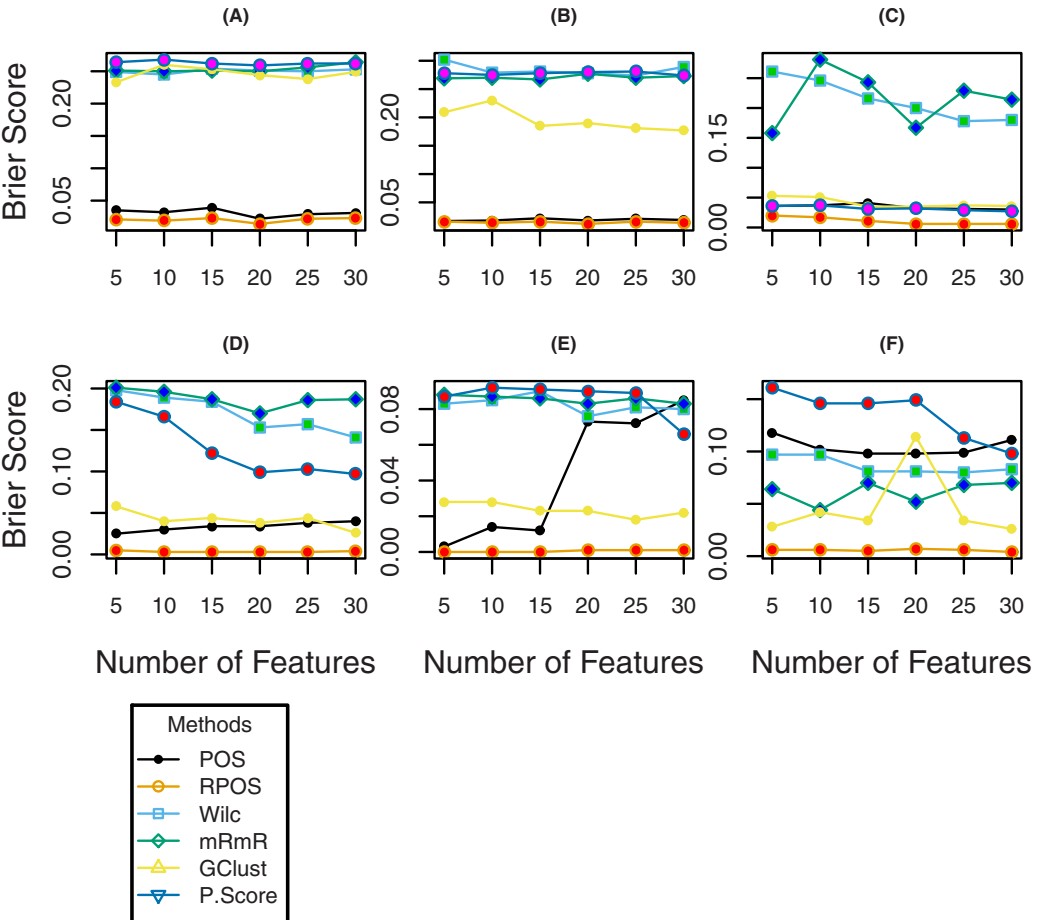

**Figure 5  Brier scores of the methods for different number of genes for the datasets; (A) TumorC, (B) Breast, (C) srbct, (D) DLBCL, (E) Lung and (F) Leukaemia.**

number of genes are given in Figs. 4, 5 and 6 respectively. From Fig. 4 it is clear that for the datasets "Breast", "DLBCL" and "Lung", the classification error rate of the proposed method RPOS is less than all the other methods for various number genes. For "TumorC" dataset the classification error of the method, i.e., Wilcoxon rank-sum test is less than all the other methods for the number of genes 5, 10, and 15 while it increases as the number of genes increases. For the remaining set of genes, the proposed method RPOS performs better than all the other methods. A similar pattern of classification error rates can be seen for the dataset "Srbct". In the case of "Leukeamia" dataset, the performance of the proposed method RPOS and the method POS for the number of genes 10, 20 and 25 are almost similar. In contrast, for the remaining set of genes, the proposed method RPOS performs better than the others.

To assess the performance of the proposed methods RPOS and the remaining methods in terms of Brier score, the results are shown by the plots given in Fig. 5, where it is clear that the proposed method RPOS outperforms all the other methods for a various number of genes. Figure 6 are the plots of the sensitivity of the proposed method RPOS and the rest of the methods. It is evident from the figure that for the datasets "TumorC",

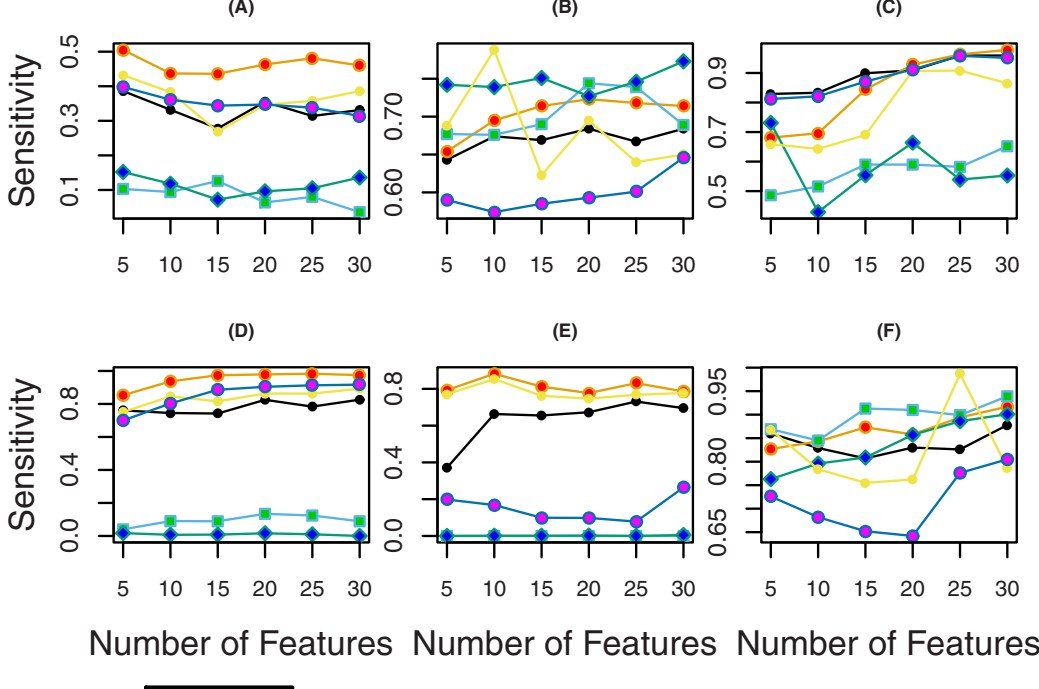

**Figure 6 Sensitivity of the methods for different number of genes for the datasets; (A) TumorC, (B) Breast, (C) srbct, (D) DLBCL, (E) Lung and (F) Leukaemia.**

"DLBCL" and "Lung" the sensitivity of the proposed method RPOS is higher than the rest of the methods for various number of genes. For the dataset "Breast" the sensitivity of the method, i.e., mRmR is more elevated in almost all the cases except the number of genes 20, where the method Wilcoxon performs better than all the other methods. In case of the dataset "Srbct" POS and sigF methods give higher sensitivity than the other methods. Wilcoxon rank-sum test outperforms the remaining methods in the case of "Leukaemia" dataset. Overall the proposed method RPOS outperforms all the other methods in 4 out of 7 datasets in terms of the performance metric, i.e., classification error rate and provides comparable results in the remaining three datasets. In terms of the performance metric, i.e., Brier scores of the proposed method RPOS outperforms all the other methods in all the seven datasets considered. In terms of sensitivity, the proposed method RPOS outperforms the rest of the methods in 4 out of 7 datasets, while gives comparable results on the remaining three datasets.

The primary aim of this research article was to devise a gene selection method to improve classification performance of machine learning algorithms on high dimensional microarray gene expression datasets. We, however, provide indices of the top 10

selected genes by our proposed method for two of the datasets, i.e., leukemia and breast. This is done for readers who might want to further assess the biological significance of the selected genes by our proposed method. Indices of the genes selected for the Leukemia dataset are (15,29,38,48,312,338,459,573,760,4847) while those of Breast dataset are (346,1481,1726,1873,2942,3259,3857,4067,4174,4435). Based on the top 10 genes selected by RPOS, we achieved 95.4% classification accuracy via SVM classifier for Leukemia dataset, and for the Breast dataset the accuracy achieved is 98.6%. Studies based on biological significance of the genes for the two datasets are given in *Kuang et al. (2010)*, *Chen & Lin (2011)*, *Bhojwani et al. (2008)*, *Castillo et al. (2019)*, *Savitsky et al. (1995)*, *Beckman et al. (1999)*.

## CONCLUSION

This paper has presented the idea of gene selection for microarray datasets via proportional overlapping analysis with the help of a more robust measure of dispersion, i.e., median absolute deviation (MAD). The core intervals of the classes in the binary class problems are constructed in a robust manner so as to minimize the effect of outliers present in the gene expression datasets in conjunction with the minimum subset of genes selected via greedy search approach. The genes having the smallest RPOS score are considered as the most discriminative, because they will have no or minimum overlapping region between the binary classes. The relative dominant class (RDC) for each gene is also calculated. Genes in the relative dominant class are arranged according to an increasing order of RPOS scores. This forms two mutually exclusive groups of genes based on RDC and RPOS scores. The genes are arranged according to RDC and RPOS scores in a round robin-fashion to develop a gene ranking list. These ranked genes do not contain the genes selected via greedy search approach. The final set of genes is selected by combining the chosen genes via greedy search approach, and the topmost ranked genes in the genes ranking list. The dimension of datasets is then reduced by including the selected genes only and discarding the rest. Classification methods; random forest, support vector machine and $k$-nearest neighbour methods have been used to assess the performance of the proposed method in comparison with other widely used gene selection methods.

The results of the proposed method indicate that it performs better in terms of almost all the performance metrics considered, i.e., classification error rate, Brier score and sensitivity. The efficiency of the proposed method is also supported by constructing boxplots for the error rate. Furthermore, the stability of the proposed method is also assessed for various number of genes. The results show that the proposed method is more stable for varying number of genes as compared to the rest of the methods.

The reason for selecting the most discriminative gene for binary classification by the proposed method is that the core intervals of the classes are constructed by the more robust measure of dispersion, i.e., median absolute deviation (MAD) than the measure of the interquartile range (IQR) used in *Mahmoud et al. (2014)*. Moreover, the breakdown point of MAD is 50% while that of IQR is 25%, which make the former less vulnerable to the outliers present in the gene expression datasets.

For future work in the direction of the current study, one could use the robust measures of dispersions like $Q_n$ and $S_n$ statistics rather than median absolute deviation (MAD). This study can be extended to multiclass problems as well. Moreover, one could use this technique in situations where the response variable is continuous.

Although this method is efficient and selects the most discriminative genes, however, there is still the possibility that two (or more) genes selected in the final set might be similar. This could cause the problem of redundancy in the selected set. One of the possible ways to eliminate this problem is to use the Least Absolute Shrinkage and Selection Operator (LASSO) method in conjunction with the proposed method. Another way to deal with this issue is to divide the entire set of features into a set of clusters and then apply the proposed method on each cluster (*Khan et al., 2019*; *Shamsara & Shamsara, 2020*; *Sharbaf, Mosafer & Moattar, 2016*). The final set of genes, in that case, will be the combination of genes selected from all the clusters. Extending performance assessment of selected genes to other recent classification methods (*Khan et al., 2020a*, *Gul et al., 2018*; *Khanal et al., 2020*; *Khan et al., 2020b*) could further validate the proposed gene selection methods.

### Funding
The authors received no funding for this work.

### Competing Interests
The authors declare that they have no competing interests.

### Author Contributions
- Muhammad Hamraz conceived and designed the experiments, performed the experiments, analyzed the data, performed the computation work, prepared figures and/or tables, and approved the final draft.
- Naz Gul analyzed the data, prepared figures and/or tables, and approved the final draft.
- Mushtaq Raza performed the computation work, authored or reviewed drafts of the paper, and approved the final draft.
- Dost Muhammad Khan performed the computation work, prepared figures and/or tables, authored or reviewed drafts of the paper, overall supervision, and approved the final draft.
- Umair Khalil performed the computation work, authored or reviewed drafts of the paper, and approved the final draft.
- Seema Zubair performed the computation work, authored or reviewed drafts of the paper, and approved the final draft.
- Zardad Khan performed the computation work, prepared figures and/or tables, authored or reviewed drafts of the paper, overall supervision, and approved the final draft.

## Data Availability

Leukemia data is available at CRAN: https://cran.r-project.org/web/packages/propOverlap/index.html. The data can be loaded in R by installing and loading the R library propOverlap and then using the command data (leukemia).

nki data is available at CRAN: https://cran.r-project.org/web/packages/penalized/index.html

The data can be loaded in R by installing and loading the R library penalized and then using the command data (nki70).

Colon data is available at: https://www.openml.org/d/1432.

Breast data is available at the following:
- http://llmpp.nih.gov/DLBCL/ (accessed Oct 29, 2003).
- http://www.stjuderesearch.org/data/ALL1 (accessed Nov 5, 2003).
- http://www.ncbi.nlm.nih.gov/geo (accessed Nov 5, 2003).
- http://www.broad.mit.edu/cgi-bin/cancer/datasets.cgi (accessed Nov 5, 2003).
- http://www.broad.mit.edu/cgi-bin/cancer/datasets.cgi (accessed Nov 5, 2003).
- http://portals.broadinstitute.org/cgi-bin/cancer/datasets.cgi.

The GSE4045 data is available at NCBI: 200004045.

https://www.ncbi.nlm.nih.gov/gds/?term=GSE4045.

Prostate data is available at https://www.openml.org/search?q=AP_Breast_Prostate&type=data

Srbct data is available at CRAN: https://file.biolab.si/biolab/supp/bi-cancer/projections/info/SRBCT.html.

Lung data is available at: https://file.biolab.si/biolab/supp/bi-cancer/projections/info/lung.html.

DLBCL data is available at: https://file.biolab.si/biolab/supp/bi-cancer/projections/info/DLBCL.html.

TumorC data is available at: https://www.openml.org/d/1107.

## Supplemental Information

Supplemental information for this article can be found online at http://dx.doi.org/10.7717/peerj-cs.562#supplemental-information.

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
