# Peer review of "Robust proportional overlapping analysis for feature selection in binary classification within functional genomic experiments"

_PeerJ Computer Science, doi:10.7717/peerj-cs.562_

## Round 0.1 · original submission · Major Revisions

Careful proofreading is necessary.

Reviewer 1 ·

Basic reporting

(1) The manuscript needs English proofreading.

Experimental design

(1) several data are unbalanced data. So, it is better to use g-mean instead of classification accuracy.
(2) The authors need to specify how they tune the SVM parameters.

Validity of the findings

Non.

Additional comments

Need to add several recent papers, such as
https://doi.org/10.1007/s11634-018-0334-1

Reviewer 2 ·

Basic reporting

The whole manuscript needs English expert for editing.

Experimental design

For the SVM hyperparameters, the authors need to specify how they choose them.

Validity of the findings

Ok.

Additional comments

Non.

Reviewer 3 ·

Basic reporting

The authors of the paper titled " Robust Proportional Overlapping Analysis for Feature Selection in Binary Classification within Functional Genomic Experiments" have proposed a new feature selection method. The authors have done various analysis to establish their points. I think the authors should address my queries to make this paper accepted.

Experimental design

line 194: Why cross-validation is not considered here?

line 196: Why 500 runs are considered, any specific reason for this? What is the reason behind choosing Random forest (RF), support vector machine (SVM), and k-Nearest neighbours?

Table 1: the gene count for nki dataset is very low, not acceptable for classification. The class sizes (96/48) is not clear to me, can you please explain them.

line 303: If there is a possibility of improving the method as you have mentioned in the last para before the Reference section. Then why have not you used that in this paper?

Validity of the findings

line 222: Do provide the selected gene names and give their biological significance? Why they provide better classification results please justify that?

Table 4: Why maximum genes count are 30?

line 294: ''median absolute deviation (MAD) than the measure of the interquartile range (IQR) used in ?''. Reason for ?

Additional comments

The references are very old please update them with the latest references.

The authors should read the entire manuscript carefully and improve the quality of the language. Lots of typo errors are exist.

According to me, the paper has lacked novelty. The latest related research discussion is required to judge the performance of the proposed feature selection method.

---

## Round 0.2 · accepted · Accept

The authors have addressed the reviewer comments.

Reviewer 1 ·

Basic reporting

The authors have adequately addressed all the comments that raised in a previous round of review.

Experimental design

non.

Validity of the findings

non.

Additional comments

The authors have adequately addressed all the comments that raised in a previous round of review.

Reviewer 2 ·

Basic reporting

The manuscript needs English proofreading.

Experimental design

non.

Validity of the findings

non.

Additional comments

The manuscript needs English proofreading.